# HuR (ELAVL1) Stabilizes *SOX9* mRNA and Promotes Migration and Invasion in Breast Cancer Cells

**DOI:** 10.3390/cancers16020384

**Published:** 2024-01-16

**Authors:** Jesús Morillo-Bernal, Patricia Pizarro-García, Gema Moreno-Bueno, Amparo Cano, María J. Mazón, Pilar Eraso, Francisco Portillo

**Affiliations:** 1Departamento de Bioquímica UAM, Instituto de Investigaciones Biomédicas Sols-Morreale, CSIC-UAM, 28029 Madrid, Spain; jmorillo@iib.uam.es (J.M.-B.); patripizgar@gmail.com (P.P.-G.); gmoreno@iib.uam.es (G.M.-B.); ampacano19@gmail.com (A.C.); mazonmaria@gmail.com (M.J.M.); peraso@iib.uam.es (P.E.); 2Instituto Ramón y Cajal de Investigación Sanitaria (IRYCIS), 28029 Madrid, Spain; 3Fundación MD Anderson Internacional, 28033 Madrid, Spain; 4Instituto de Investigación Sanitaria del Hospital Universitario La Paz-IdiPAZ, 28029 Madrid, Spain; 5Centro de Investigación Biomédica en Red, Área de Cáncer (CIBERONC), Instituto de Salud Carlos III, 28029 Madrid, Spain

**Keywords:** HuR, SOX9, mRNA stability, migration, invasion, breast cancer cells

## Abstract

**Simple Summary:**

RNA-binding proteins (RBPs) play a critical role in controlling gene expression post-transcriptionally and their dysregulation can lead to various diseases, including cancer. One such RBP is HuR, whose levels have been associated with a poor clinical prognosis in breast cancer. Nevertheless, the precise molecular mechanism underlying this connection remains incompletely characterized. Our study uncovers that HuR targets *SOX9* mRNA in breast cancer cells and provides compelling evidence supporting HuR’s involvement in cell migration and invasion.

**Abstract:**

RNA-binding proteins play diverse roles in cancer, influencing various facets of the disease, including proliferation, apoptosis, angiogenesis, senescence, invasion, epithelial–mesenchymal transition (EMT), and metastasis. HuR, a known RBP, is recognized for stabilizing mRNAs containing AU-rich elements (AREs), although its complete repertoire of mRNA targets remains undefined. Through a bioinformatics analysis of the gene expression profile of the Hs578T basal-like triple-negative breast cancer cell line with silenced HuR, we have identified SOX9 as a potential HuR-regulated target. SOX9 is a transcription factor involved in promoting EMT, metastasis, survival, and the maintenance of cancer stem cells (CSCs) in triple-negative breast cancer. Ribonucleoprotein immunoprecipitation assays confirm a direct interaction between HuR and *SOX9* mRNA. The half-life of *SOX9* mRNA and the levels of SOX9 protein decreased in cells lacking HuR. Cells silenced for HuR exhibit reduced migration and invasion compared to control cells, a phenotype similar to that described for SOX9-silenced cells.

## 1. Introduction

RNA-binding proteins (RBPs) are indispensable regulators of gene expression at the post-transcriptional level, modulating various aspects of RNA metabolism, which include transcription, splicing, capping, polyadenylation, transport, location, translation, and turnover. Consequently, it is not surprising that a dysfunction in RBPs can impact a broad spectrum of transcripts, leading to various diseases [1,2,3], notably cancer [4,5,6,7,8].

Human antigen R (HuR), encoded by the embryonic lethal abnormal vision-like 1 (*ELAVL1*) gene, is a member of the RBP gene family ELAVL, along with HuB, HuC, and HuD. While other protein members of the ELAVL family are expressed in neurons, HuR is ubiquitously expressed in human tissues [9]. HuR primarily targets mRNAs containing AU-rich elements (AREs) in their 3′ UTR. AREs are cis-acting determinants of mRNA decay, and HuR acts as an mRNA stability factor by binding to these AREs sequences [10]. However, HuR can also influence mRNA translation [11]. Thousands of mRNAs possess HuR binding sites, with some of them identified as bona fide HuR targets, including mRNAs encoding cell cycle regulators, pro-survival proteins, and angiogenic factors, many of which are implicated in cancer [12,13].

One key process that enables carcinoma cells to acquire phenotypic plasticity and escape from the primary tumor is the Epithelial to Mesenchymal Transition (EMT). The EMT is a genetic and cellular program that results in the loss of epithelial characteristics and the acquisition of mesenchymal traits, giving rise to cells with increased mobility, migration, and invasion potential [14,15,16]. Over the last decades, several transcription factors (TFs) known as EMT inducers (EMT-TFs) have been identified. Multiple signaling pathways converge to activate one or more of the core EMT-TFs, including the zinc finger TFs SNAI1, SNAI2, ZEB1, and ZEB2, and well as the basic helix-loop-helix TFs TCF3 and TWIST1 [17,18,19]. These EMT-TFs can either bind directly to DNA or form transcriptional regulatory complexes to orchestrate the EMT by controlling the expression of numerous genes. One of the EMT’s hallmarks is the functional loss of E-cadherin (*CDH1*), an invasion suppressor protein involved in cell–cell adhesion at adherens junctions, and EMT also negatively impacts on cell polarity [20,21].

In lung adenocarcinoma, HuR forms a complex with the long non-coding RNA LINC00152, favoring the stabilization of *SNAI1*, *SNAI2*, and *ZEB1* mRNAs, thereby promoting EMT [22]. In pancreatic ductal adenocarcinoma, HuR promotes EMT through the regulation of SNAI1 and YAP1 [23,24]. In non-metastatic MCF7 breast cancer cells, HuR, together with the long non-coding RNA MALAT, forms a chromatin regulatory complex. The HuR/MALAT1 complex binds to the promoter region of the cancer stem cell marker *CD133*, repressing its expression and thus suppressing EMT [25]. However, whether HuR plays a positive role in the progression of breast cancer is presently unknown.

In this study, we aimed to investigate the role of HuR in breast cancer and identify its potential targets involved in EMT. To achieve this, we performed an RNA-Seq analysis of the expression profile of a Hs578T basal-like triple-negative breast cancer cell with silenced HuR and selected the downregulated genes. We hypothesized that for some of these genes, the decreased levels could be attributed to reduced mRNA stability caused by the absence of HuR. Subsequently, we mapped the binding sites of HuR to mRNAs with predicted AREs in their 3′ UTR. We found that in 20 mRNAs, the HuR binding sites matched ARE sequences. Among these, we selected SRY-Box Transcription Factor 9 (*SOX9*) for further study due to its established involvement in cancer [26,27,28], its causation of EMT in various cancer cell line models [29], its essential role in the survival and metastasis of triple-negative breast cancer cells [30], and its function as a negative regulator of E-cadherin expression [31,32]. Biochemical analyses confirmed that HuR binds to and stabilizes *SOX9* mRNA. Finally, we demonstrated that silencing HuR negatively affects the migration and invasion abilities of Hs578T and BT549 human basal-like triple-negative breast cancer cells in a way that mimics the phenotype of *SOX9*-silenced cells.

## 2. Materials and Methods

### 2.1. Cell Culture and Plasmids

Hs578T and BT549 human basal-like triple-negative breast cancer cell derivatives were obtained from the American Type Culture Collection. Cells were grown in Dulbecco’s modified Eagle’s medium (Merck, Darmstadt, Germany) supplemented with 10% fetal bovine serum (Gibco, Grand Island, NY, USA), 20 mM L-glutamine (Gibco), 10 U/mL penicillin (Gibco), 10 μg/mL streptomycin (Gibco), 2.5 μg/mL amphotericin B (Gibco Fungizone™), and 1 μg/mL puromycin (Merck,) or 1 μg/mL G418 (Merck) to select transfected cell lines. Cells were grown at 37 °C in a humidified 5% CO_2_ atmosphere. Cells were routinely tested for *Mycoplasma* contamination.

HuR was silenced in Hs578T and BT549 cell lines using lentiviral particles that encoded selected shRNAs (TRCN0000276129 referred to as shHuR-1 and TRCN0000276126 referred to as shHuR-2) from the MISSION TRC shRNA human library (Merck) along with a corresponding control that expressed a scrambled shRNA (shSCR). The *SOX9* 3′ UTR reporter (HmiT127361-MT06) and its corresponding control (Cmi0000001-MT06) plasmids were purchased from GeneCopoeia (Rockville, MD, USA).

### 2.2. RNA-Seq Analysis

RNA from three independent clones of shSCR and shHuR-1 cells was used to perform RNA-Seq analysis conducted by Sistemas Genómicos (Valencia, Spain). Significantly differentially expressed genes (DEGs) were selected using a cutoff of *p*-value adjusted by FDR ≤ 0.05 and a fold change greater than 2. Gene Set Enrichment Analysis (GSEA) version 4.1.0 was applied to determine which gene sets from the Molecular Signatures Database Hallmark collections were enriched among DEGs ranked by statistical significance [33]. Genes with predicted AREs in the 3′UTR along with the sequence and localization of the AREs on cDNA were extracted from the ARED PLUS Database (https://brp.kfshrc.edu.sa/ARED, accessed on 15 October 2021) [34]. HuR targets and binding sites were obtained from the ENCORI database, the Encyclopedia of RNA Interactomes, which is a compendium of interaction binding sites of RBPs on RNA derived from CLIP-seq data (https://rnasysu.com/encori/, accessed on 22 March 2022) [35]. Only HuR binding sites described in ≥5 CLIP datasets (strict stringency) were selected. The genomic location and sequences of HuR binding sites were viewed using UCSC Integrated Genome (https://genome.ucsc.edu, accessed on 29 April 2022) [36] and the binding sites were manually mapped on to cDNA sequences extracted from Ensembl Genome Browser version GRch37 (https://ensembl.org/, accessed on 29 April 2022).

### 2.3. RNA Extraction, cDNA Synthesis and Quantitative PCR (qPCR)

RNA extraction was performed using a QIAcube Nucleic Acids automatic extraction system (Qiagen, Hilden, Germany) and RNA purity was assessed using a 2100 bioanalyzer instrument (Agilent, Santa Clara, CA, USA). These procedures were carried out at the Genomics Core Facility at the Instituto de Investigaciones Biomédicas Sols-Morreale CSIC-UAM (Madrid, Spain). Reverse transcription of 1 μg of RNA was performed using 200 units of M-MLV reverse transcriptase (Promega Corporation, Madison, WI, USA), 5 μL M-MVL buffer (Promega), 0.5 μg random primers (Promega), 10 mM dNTP mix (Bioron, Römerberg, Germany), and 25 units of RNaseOUT™ recombinant ribonuclease inhibitor (Invitrogen, Waltham, MA, USA) in a final reaction volume of 25 mL. Real-time qPCR was performed using a power SYBR green master mix (Thermo Fisher Waltham, MA, USA) on an Applied Biosystems StepOne™ machine (Thermo Fisher). Each reaction was performed with 20 ng of cDNA and 9 pmol of specific forward and reverse primers (Appendix A). Values were normalized to the levels of *GAPDH* mRNA, encoding the housekeeping protein GAPDH, and relative expression levels were analyzed using the 2^−ΔΔCt^ method. qPCRs were conducted in three independent samples assayed in triplicate.

### 2.4. mRNA Stability

To measure mRNA stability, cells were treated with 5 μg/mL actinomycin D (Sigma-Aldrich, San Luis, MI, USA) at the indicated times. Total RNA was purified and mRNAs half-life analyzed using RT-qPCR. Transcript levels were normalized to 18S rRNA. This experiment was carried out in three biological replicates assayed in triplicate.

### 2.5. Ribonucleoprotein Immunoprecipitation Assay (RIP) 

The RIP assay was performed using a Magna RIP™ kit (Millipore, Burlington, MA, USA) following the manufacturer’s protocol. Briefly, Hs578T-shSCR cells were collected and suspended in an RIP lysis buffer. Cell extracts were incubated overnight at 4 °C with magnetic beads protein A/G conjugated with 5 μg of anti-HuR (Millipore, cat. #03-102), or IgG antibody (Millipore; cat. #PP64B). Each immunoprecipitate (IP) was treated with 0.5 mg/mL proteinase K (15 min at 55 °C). RNA isolated from IP material was analyzed using RT-qPCR. *GAPDH* mRNA in each IP was used for normalization of RIP results. Actin (*ACTB*) mRNA levels were used as a positive control. The RIP assay was performed in three independent replicates assayed in triplicate.

### 2.6. RNA Pull Down of SOX9 ARE 

RNA pull down was performed as described [37]. A customized biotinylated RNA probe corresponding to the *SOX9* ARE element or a mutated version was used. Sequences are listed in Appendix A. RNA probes (50 pmoles) were incubated with 500 μg of cell lysate and incubated for 2 h at room temperature (RT) in a buffer containing 10 mM Tris-HCl pH 8, 1 mM EDTA, 250 mM NaCl, 0.5% TritonX-100 buffer, and protease (Roche) and RNase inhibitors (Ribolock, Thermo Fisher). Probes were captured with 500 μg of Dynabeads M-280 streptavidin for 2 h at RT. Beads were washed extensively in the same buffer prior to addition of Laemmli’s buffer and Western blotting.

### 2.7. 3′ UTR Luciferase Reporter Assays

Hs578T shSCR and shHuR cells were transfected with 1 μg of *SOX9* 3′ UTR reporter and corresponding control plasmids using Lipofectamine 2000 (Life Technologies, Carlsbad, CA, USA). Luciferase activity was measured using a Luc-Pair™ Duo-Luciferase Assay Kit 2.0 (GeneCopoiea). Firefly luciferase activity was normalized to Renilla luciferase activity and plotted as a percentage of the control cells (shSCR). Experiments were performed in triplicate and repeated at least three times.

### 2.8. Western Blot Analysis

Protein samples were resolved on 10% SDS-PAGE gel and transferred to PVDF membranes (Millipore). The membrane was blocked with 5% non-fat milk in TBST for 1 h at RT. After blocking, the membrane was incubated overnight at 4 °C with antibodies against HuR (ab200342, Abcam, Cambridge, UK), HNRNPD (sc-166577, Santa Cruz Dallas, TX, USA), PABPC1 (NB120-6125SS, Novus, Littleton, CO, USA) SOX9 (AB5535, Millipore), or GAPDH (CB1001-500, Millipore). Uncropped Western blot images can be found in Appendix A–S7.

### 2.9. Migration and Invasion Assays

Cell migration was analyzed using wound-healing assays performed essentially as described [38]. Briefly, cells were treated for 2 h with 10 μg/mL mitomycin C in medium with 10% FBS to inhibit proliferation. After treatment, monolayers (90% confluent cells) were scratched using a 10 µL pipette tip. Images of the wound width were captured at the indicated times and quantified using ImageJ software 1.53f. Invasion was analyzed in Transwell assays using Corning BioCoat matrigel invasion chamber (Thermo Fisher). The 2 × 10^4^ cells were suspended in culture medium with 0.2% FBS and placed in the upper chamber. The lower chamber contained 0.75 mL of medium with 20% FBS as a chemoattractant. Cells were allowed to invade for 24 h at 37 °C and 5% CO_2_. Non- invading cells in the upper chamber were removed with a cotton swab and membranes were fixed in 4% formaldehyde in PBS and stained with crystal violet. The total number of cells invading the lower surface was counted using ImageJ software. Eight fields for each condition were quantified using ImageJ software. Three independent replicates were analyzed.

### 2.10. Statistical Analysis

The quantitative data are expressed as mean ± standard error of the mean (SEM). Statistical differences were calculated using unpaired two-tailed Student’s *t* test. *p*-values ≤ 0.05 were considered as statistically significant. * *p* < 0.05; ** *p* < 0.01; *** *p* < 0.001; ns, not significant.

## 3. Results

### 3.1. Gene Expression Profile of HuR-Silenced Cells

To identify target mRNAs regulated by HuR, we conducted an RNA-Seq analysis of the gene expression profile in Hs578T cells with silenced HuR (shHuR-1) compared to control cells (shSCR). The effectiveness of HuR silencing was confirmed using RT-qPCR and Western blot (Appendix A). The RNA-Seq analysis (Appendix A) revealed a total of 1140 DEGs with a log2 fold change of ≥1 or ≤−1, with the majority being downregulated in shHuR cells (797 out of 1140 DEGs) (Figure 1A). The RNA-Seq data for downregulated DEGs were validated using RT-qPCR analysis of randomly selected DEGs in shHuR-1 and shHuR-2 cells compared to shSCR cells (Figure 1B). Gene Set Enrichment Analysis (GSEA) of the RNA-Seq data revealed enrichment of several pathways, with the EMT pathway being the most significantly enriched signature (Figure 1C).

Interestingly, *CDH1* expression was upregulated two-fold in shHuR cells but none of the classic EMT-TFs (SNAI1, SNAI2, TCF3, TWIST1, ZEB1, and ZEB2) appeared to be deregulated in shHuR cells (Appendix A).

### 3.2. Selection of Putative HuR Targets

Given that HuR primarily protects its target mRNAs from degradation by binding to AREs located in the 3′UTR of the mRNA, we established three criteria for selecting putative HuR mRNA targets: (1) significant downregulation in the gene expression profile of shHuR cells, (2) presence of AREs in the 3′ UTR, and (3) overlap between HuR binding sites and ARE. Using a log2 fold change threshold of <−1.5, we identified 535 downregulated DEGs (Appendix A). Among these, we found 188 DEGs with AREs in the 3′ UTR region using the AU-Rich Element Database. Subsequently, we searched the ENCORI database for HuR binding sites in the 3′ UTR of the selected DEGs and identified 63. Finally, we manually mapped the HuR binding site sequences extracted from the UCSC Integrated Genome database to the cDNA 3′ UTR and found that 20 DEGs met all the criteria. The details of this selection process are provided in Appendix A. Figure 2A displays the coordinates of the ARE and HuR binding sites in the cDNA of the selected genes, and Appendix A presents a schematic view of the position of the ARE and HuR binding sites. Several well-established HuR target mRNAs including *CCL2*, *CXCL8*, and *PTGS2* [39,40] were among the putative targets. We analyzed the expression levels of some of the selected putative targets in Hs578T cells with HuR silenced (shHuR-1, shHuR-2) compared to control cells (shSCR) using RT-qPCR (Figure 2B). Furthermore, we extended this analysis to BT549, another human basal-like triple-negative breast cancer cell line. HuR silencing was confirmed using RT-qPCR (Appendix A), and the expression levels of HuR putative targets were evaluated using RT-qPCR (Appendix A). Comparable downregulation of the HuR putative targets was observed in Hs578T and BT549 HuR-silenced cells.

One of the selected DEGs of particular interest was *SOX9*, coded for a transcription factor that plays a crucial role in the development of various organs and as a regulator of stem cells [26]. *SOX9* can act as either a proto-oncogene or a tumor suppressor gene depending on the type of cancer [29]. In breast cancer, SOX9 is involved in promoting EMT, metastasis, survival, drug resistance, stem cell maintenance, immune evasion, and modulation of the tumor microenvironment. Additionally, high expression of SOX9 is associated with a poorer prognosis for patients with breast cancer [30]. Therefore, we focused on *SOX9* as a potential target for HuR in the present study.

### 3.3. HuR Binds to SOX9 mRNA

Downregulation of SOX9 expression in HuR-silenced cells was confirmed by analyzing the levels of *SOX9* mRNA in shHuR-1 and shHuR-2 cells compared to shSCR cells using RT-qPCR (Figure 3A). The binding of HuR to the mRNA of *SOX9* was assessed using RIP assay in shSCR cells, using an anti-HuR antibody and rabbit IgG as a control. Previously, the endogenous expression of HuR and the efficiency of immunoprecipitation of this protein were evaluated by Western blot. HuR was present in cell extracts and immunoprecipitated with the anti-HuR antibody (Figure 3B, upper panel). After isolating the RNA present in HuR–RNP complexes, the mRNA levels of the HuR control target *ACTB* were measured using RT-qPCR analysis to test the success of the RIP assay. Subsequently, the levels of *SOX9* mRNA present in the immunoprecipitated HuR–RNP complexes were measured. The results (Figure 3B, lower panel) showed a robust enrichment of both mRNAs in the immunoprecipitated HuR–RNP complexes relative to the levels in the IP samples obtained using a control antibody (IgG). This result indicates that HuR binds to the mRNA of SOX9.

Next, we decided to test whether HuR binds to the *SOX9* ARE sequence. To this end, we performed an RNA pull down experiment using shSCR cells and a biotinylated RNA probe corresponding to the *SOX9* ARE or a mutated version. Results showed a strong binding of HuR to the *SOX9* wt ARE probe but not to the mutated probe (Figure 3C). As negative controls, we also tested HNRNPD (AUF1) which does not present described *SOX9* mRNA binding sites, and PABPC1 (Poly(A) Binding Protein Cytoplasmic 1) which binds to the polyA tail of mRNAs which, as expected, were only detected in the input fraction.

Taken together, these results demonstrate a direct interaction between HuR and the ARE present in the 3′ UTR of *SOX9* mRNA, although they do not rule out the possibility that HuR may bind to other sequences. In fact, our analysis of the CLIP-seq experiment database reveals at least 27 HuR CLIP-peaks in the *SOX9* 3′UTR region (Appendix A).

### 3.4. HuR Stabilizes SOX9 mRNA

Binding of HuR to AREs located in the 3′ UTR regulates gene expression by increasing mRNA stability. Thus, we examined whether HuR silencing influences the activity of a luciferase-*SOX9* 3′ UTR chimeric mRNA using a luciferase reporter assay. As shown in Figure 4A, HuR silencing in Hs578T cells clearly decreased luciferase activity compared to control (shSCR) cells.

To ascertain the biological consequences of the HuR binding to *SOX9* mRNA, we measured *SOX9* mRNA half-life in shHuR and control cells using an actinomycin D chase experiment. HuR silencing decreased *SOX9* mRNA half-life from 6 h detected in control (shSCR) to 2 h (Figure 4B).

To determine if the shorter half-life of *SOX9* mRNA observed in cells devoid of HuR has an impact at the protein level, we analyzed SOX9 protein levels in shSCR and shHuR cells using Western blot. As seen in Figure 4C, the SOX9 level was clearly lower in shHuR cells compared to the control. This result was confirmed in BT549 cells silenced for HuR (Appendix A).

Altogether, the results obtained so far establish that HuR binding to *SOX9* mRNA positively regulates SOX9 expression.

### 3.5. Proliferation, Migration and Invasion Are Decreased in HuR-Silenced Cells

Since SOX9 promotes proliferation, migration, and invasion in human basal-like triple-negative breast cancer cell [30], we decided to explore the consequences of HuR silencing on such parameters in Hs578T and BT549 cell lines. The proliferation of HuR-silenced cells was evaluated using MTT assay at 24, 48, 72, and 96 h where Hs578T silenced cells exhibited a significantly diminished cell viability compared to control shSCR cells (Figure 5A). The migratory capacity of the Hs578T cells devoid of HuR was analyzed in scratch-wound-healing assays, showing that shHuR cells had a lower migratory behavior than control shSCR. The entire wound surface was colonized by HuR-expressing cells 48 h after the scratch was made, whereas at this time the HuR-silenced cells had covered only 75% (shHuR-1) and 70% (shHuR-2) of the wound surface (Figure 5B). The invasive capacity of Hs578T-silenced cells was examined in matrigel invasion assays. In these experiments, HuR silencing suppressed cell invasion by nearly 60% in Hs578T cells compared to controls (Figure 5C). The effect of HuR silencing on proliferation, migration, and invasion in BT549 cells devoid of HuR compared to the shSCR controls yielded similar results (Appendix A–F).

## 4. Discussion

In the present study, we combined a gene expression analysis of HuR-silenced cells and a bioinformatics-based selection procedure to uncover putative mRNAs regulated by HuR in Hs578T human basal-like triple-negative breast cancer cells. We discovered 20 genes exhibiting HuR binding sites encompassing AREs located in the mRNA 3′UTR, which are suggestive of being HuR targets. Despite the limitations of this approach, we identified three genes (*CCL2*, *CXCL8,* and *PTGS2*) described as bona fide HuR substrates [39,40]. In this study we showed that *SOX9* mRNA is also a HuR substrate. We have not found any evident connection of HuR with the remaining selected genes in the current literature, and future research studying whether such a link exists and its functional consequences will expand the spectrum of HuR targets.

Several reports have suggested a role for HuR and SOX9 in breast cancer progression [12,25,27,28,29,30,31,32], but it was unknown if there was any interconnection between both proteins. In the present report we uncovered a functional link between both partners in basal-like triple-negative breast cancer cells. HuR binds to *SOX9* mRNA, promoting its stabilization and increasing SOX9 protein levels. The silencing of HuR negatively impacts proliferation, migration, and invasion proficiency of the cells. Altogether, our results suggest that HuR’s role in breast cancer progression could be, at least partly, mediated by the stabilization of *SOX9* mRNA.

Our results reveal a new role for HuR in the regulation of EMT in breast cancer cells. Earlier findings showed that in non-metastatic MCF7 cells (a cell model of luminal ER^+^ breast cancer), HuR, and MALAT1 form repressive chromatin complexes that impact *CD133* expression, suppressing EMT [25]. Now, we have proved that in metastatic Hs578T cells (a cell model of basal-like triple-negative breast cancer), HuR promotes cell migration and invasion. Interestingly, the expression level of MALAT1 is lower in triple-negative breast tumors than in ER^+^ tumor samples [25]. Perhaps the MALAT1–HuR axis plays a pivotal role in the regulation of EMT in breast cancer. In ER^+^ tumors, the elevated levels of MALAT1 could favor the transcriptional-associated side of HuR, while in triple-negative breast tumors, the lower levels of MALAT1 could allow the mRNA-stabilizing aspect of HuR. Future experiments aimed at evaluating whether overexpression of MALAT1 in cell models of basal-like triple-negative breast cancer reverses the stabilization of *SOX9* mRNA could help assess the role of MALAT1-HuR in EMT regulation.

In recent years, multiple mechanisms involved in the control of SOX9 gene expression have been revealed. At the transcriptional level, *SOX9* expression is controlled by many signaling pathways including transforming growth factor β, bone morphogenetic protein, fibroblast growth factor, or sonic hedgehog, among others. Another mechanism controlling *SOX9* transcription is the methylation of the promoter, as it has been shown that the canonical WNT signaling represses *SOX9* via methylation of its promoter. At the post-transcriptional level, miRNAs and other non-coding RNAs (lncRNAs and circRNAs) regulate *SOX9* expression to promote or inhibit cancer progression [41,42,43]). Our present study, showing that HuR stabilizes *SOX9* mRNA, uncovers a new mechanism regulating SOX9 expression.

It remains to be clarified which specific facet of SOX9 mRNA destabilization is counteracted by HuR. At least two alternatives can be foreseen. On one side, it is feasible that HuR blocks the access of destabilizing proteins to the SOX9 3′UTR region. Three mRNA decay-promoting RBPs have been described so far: HNRNPD (AUF1), the tristetraprolin RBP family, consisting of ZFP36, ZFP36L1, and ZFP36L2, and the KH-Type Splicing Regulatory Protein (KHSRP) [10]. An exploration of the ENCORI database revealed two binding sites for ZPF36 in the *SOX9* mRNA sequence. Nevertheless, the small number of CLIP experiments performed so far precludes us from discarding the binding of other destabilizing proteins. On the other hand, it is also possible that HuR impedes the binding of destabilizing miRNAs to the *SOX9* 3′UTR. In breast cancer, the SOX9 3′UTR is targeted by different miRNAs (miR-133b, -134, -140, -190, 215, and -224), leading to reduced SOX9 expression and operating as tumor suppressors, reducing the metastasis of breast cancer cells [42,44,45,46,47].

Further experiments are needed to determine whether HuR counteracts RBPs or miRNA binding to prevent *SOX9* mRNA decay, which will increase our knowledge in the role of HuR in the progression of breast cancer.

## 5. Conclusions

In summary, our data uncovered a new functional relationship between HuR and SOX9 in breast cancer cells. HuR binds to *SOX9* mRNA, counteracting its degradation. The silencing of HuR in breast cancer cells produces a phenotype that mimics that produced by the silencing of SOX9. This leads us to speculate that the phenotype resulting from HuR silencing may be mediated by the destabilization of SOX9 mRNA. Taken together, our findings highlight the importance of the HuR–SOX9 axis in the migration and invasion abilities of basal-like triple-negative breast cancer cells.

## Figures and Tables

**Figure 1 cancers-16-00384-f001:**
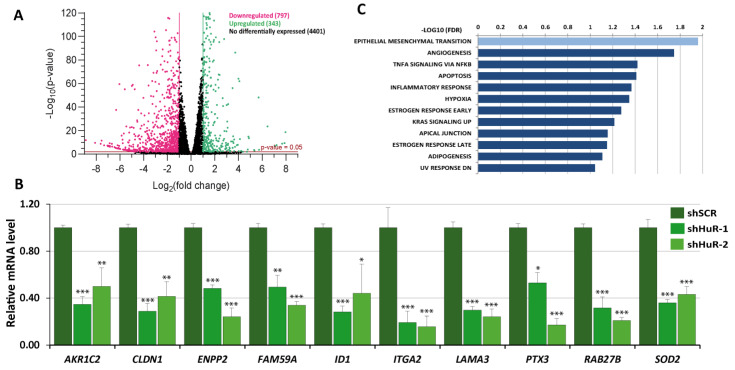
Gene expression profile of Hs578T HuR-silenced cells. (**A**) Volcano plot displaying changes in the transcriptome of HuR-depleted cells. Genes with a log2 fold change of > 1.0 or < −1.0 and *p* < 0.05 are shown as pink (downregulated) or green (upregulated) dots, respectively. (**B**) Validation of downregulation of selected DEGs in Hs578T HuR-ablated cells using quantitative RT-PCR. Data represent the mean ± SEM of three independent experiments with triplicate assays. The *p*-value was calculated using a two-sided unpaired Student’s *t*-test (* *p* < 0.05; ** *p* < 0.01; *** *p* < 0.001). (**C**) GSEA plot of DEGs showing enrichment of Hallmark signatures. The y-axis displays the FDR-values (−Log10 FDR) of the enrichments.

**Figure 2 cancers-16-00384-f002:**
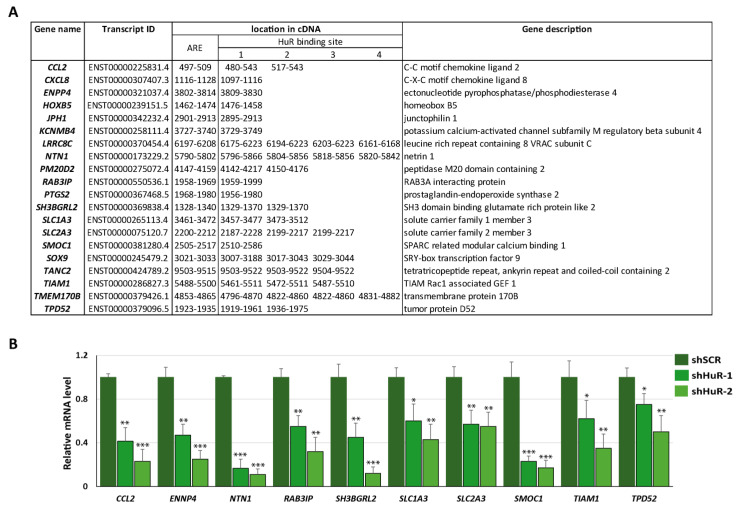
Selection of putative HuR targets in Hs578T cells. (**A**) DEGs list displaying the location of ARE and overlapping HuR binding sites in the cDNA of the selected DEGs. (**B**) Validation of downregulation of putative targets in Hs578T HuR-silenced cells using quantitative RT-PCR. Data represent the mean ± SEM of three independent experiments assayed in triplicate. The *p*-value was calculated using a two-sided unpaired Student’s *t*-test. (* *p* < 0.05; ** *p* < 0.01; *** *p* < 0.001).

**Figure 3 cancers-16-00384-f003:**
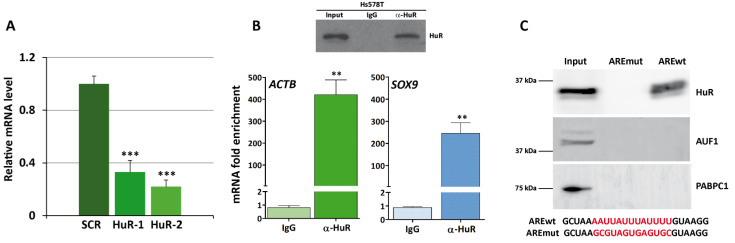
HuR binds to the SOX9 ARE in Hs578T cells. (**A**) Quantitative RT-PCR confirming downregulation of SOX9 in HuR-silenced cells. Data are the mean ± SEM of three independent experiments assayed in triplicate (*** *p* < 0.001). (**B**) RIP analysis of shSCR cells using IgG and anti-HuR antibodies. HuR immunoprecipitation efficiency was tested using Western blot (upper panel). ACTB and SOX9 mRNAs were quantified using RT-qPCR and represented as fold enrichment compared to IgG RIP analysis. Data are the mean ± SEM of three independent experiments assayed in triplicate (** *p* < 0.01). (**C**) HuR pull down using biotinylated synthetic RNA corresponding to the wild type SOX9 ARE (AREwt) or a mutated version (AREmut). The presence of HuR, hnRNPD (AUF1), or PABPC1 in the input and pull down fractions was detected using Western blot.

**Figure 4 cancers-16-00384-f004:**
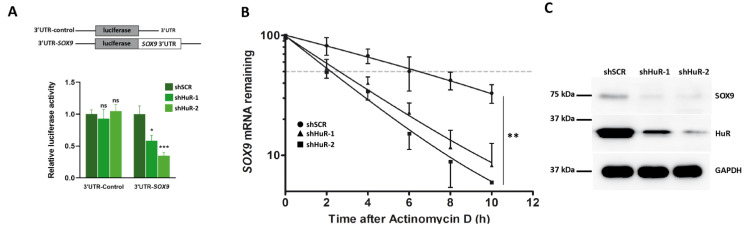
HuR silencing reduces SOX9 expression. (**A**) Relative luciferase activity in control (shSCR) or HuR-silenced Hs578T cells (shHuR-1, shHuR-2) transfected with *SOX9* 3′UTR reporter plasmids (3′UTR-SOX9) or control plasmid (3′UTR-control). Data are the mean ± SEM of three independent experiments (* *p* < 0.05; *** *p* < 0.001; ns, not significant). (**B**) *SOX9* mRNA half-life was measured using RT-qPCR in control (shSCR) or HuR-depleted (shHuR-1, shHuR-2) Hs578T cells treated with actinomycin D. The data were normalized to 18S rRNA levels and represented as a percentage of the mRNA levels measured at time 0, before adding actinomycin D, using a semilogarithmic scale. A discontinuous horizontal line indicates a 50% decrease in mRNA abundance. Data are the mean ± SEM of three independent experiments (** *p* < 0.01). (**C**) Western blot analysis of HuR and SOX9 protein levels in control (shSCR) or HuR-silenced (shHuR-1, shHuR-2) Hs578T cells. GAPDH was used as a loading control.

**Figure 5 cancers-16-00384-f005:**
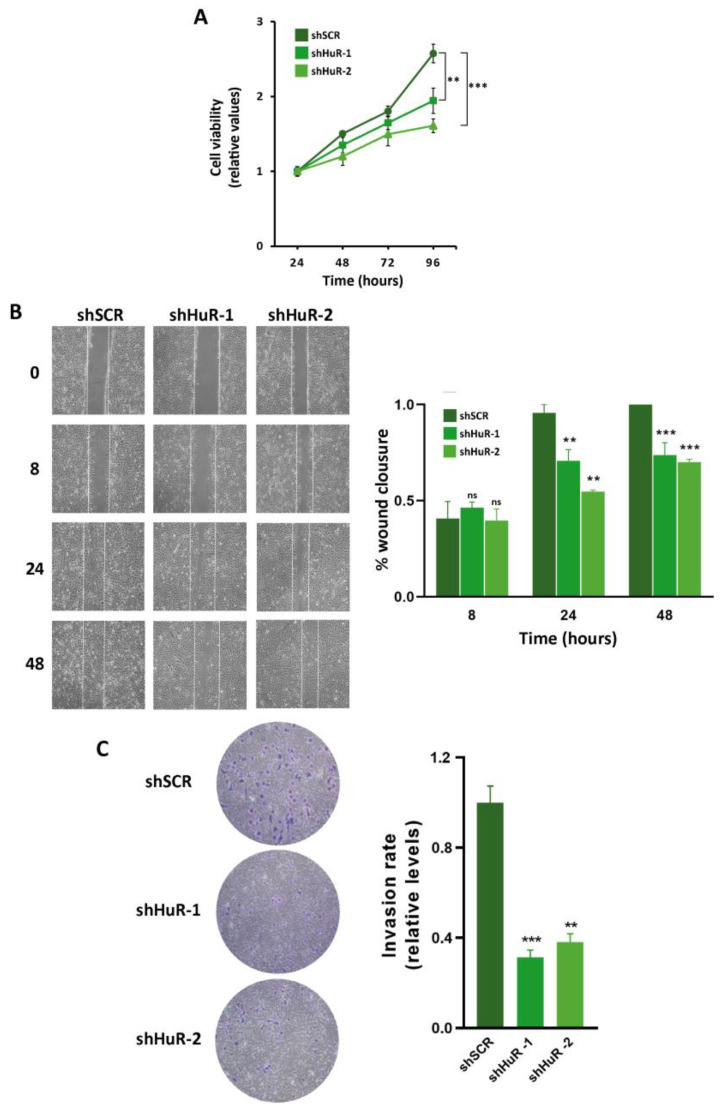
HuR silencing decreases proliferation migration and invasion in Hs578T cells. (**A**) Cell viability analyses of control (shSCR) or HuR-silenced (shHuR-1, shHuR-2) cells. MTT assays were performed at the indicated time points after seeding. Data were normalized to the value detected at 24 h. Error bars represent the mean ± SEM of three independent experiments assayed in quintuplicate (**B**) Cell motility of the indicated clones was analyzed using wound-healing assay. Images were taken at 0, 8, 24, and 48 h (h) after culture scratch (upper panel) and area closure was quantified using ImageJ (lower panel). Results represent the mean ± SEM of three independent experiments. (**C**) Cells were seeded in the upper chambers of Transwell, allowed to migrate for 18 h, membrane stained with crystal violet, and as described in Methods section were photographed. Left panel: representative images of the lower membrane (invading cells). Right panel: percentage of invasiveness by direct measurement with ImageJ. Values represent the mean ± SEM from three independent experiments performed in triplicate. (** *p* < 0.01, *** *p* < 0.001; ns, not significant). The images in (**B**) and (**C**) were taken using an inverted microscope with the 10× objective.

## Data Availability

Data are contained within the article and Appendix A.

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
