# Peer review of "HuR (ELAVL1) Stabilizes SOX9 mRNA and Promotes Migration and Invasion in Breast Cancer Cells"

_cancers, 2024, doi:10.3390/cancers16020384_

Round 1

Reviewer 1 Report

Comments and Suggestions for Authors

Morillo-Bernal and colleagues present an interesting piece of work on the role of HuR in the migration and invasion phenotypes of triple-negative breast cancer cells. SOX9 is identified as a potential HuR regulated target. RNA immunoprecipitation and pulldown experiments show an interaction between HuR and SOX9 mRNA. Mechanistically, HuR seems to stabilize SOX9 mRNA levels, contributing to its expression level. Functionally, HuR silencing reduces migration and invasion of breast cancer cells. The work seems sound and proper controls are in place. Specific comments follow.

Specific comments:

Introduction section, lines 65 to 67: there are additional examples of HuR-mediated regulation of EMT features in breast cancer and in other cancer types: Latorre et al. 2016, Cancer Research (DOI: 10.1158/0008-5472.CAN-15-2018); Xu et al. 2022, Cell Death and Disease (DOI: 10.1038/s41419-022-05164-2). This paragraph should be extended;

Results section, lines 193 and 194: “The RNA-Seq analysis revealed a total of 5,541 significantly expressed genes (FDR-adjusted p-value ≤ 0.05)…”. This sentence is not informative. Should be removed for clarity, and to give focus only to the differentially expressed genes (DEGs) between experimental conditions, which are the ones that really matter;

Results section, related with Figure 1C: please provide Gene Ontology and KEGG pathway analysis on the list of DEGs, separating the downregulated from the upregulated class;

Results section, related with Figure 1B: given the putative association with EMT, Authors should validate EMT-related targets from the RNA-seq data. Please provide data regarding E-Cadherin and EMT-TFs (SNAI1, SNAI2, TCF3, TWIST1, ZEB1 and ZEB2) protein expression levels by western blot to assess effect of HuR inhibition on EMT targets;

Results section, related with Figure 3C: for the HuR pulldown experiment, a biotinylated synthetic RNA corresponding to a wild-type ARE (AREwt) or a mutated version (AREmut) were used. But the oligos are quite short. Hence, HuR might be simply recognizing the ARE element, irrespective of the sequence context. This experiment would be more informative using an extended sequence for the pulldown in order to be sure HuR is directly binding to the ARE within the SOX9 mRNA;

Results section, related with Figure 5: a key experiment is lacking concerning the functional effect of SOX9 mRNA destabilization by HuR inhibtion. Authors must perform functional rescue experiments by overexpressing SOX9 in shHuR cells and evaluate the effect on viability, migration and invasion to determine if the observed phenotypes are mainly mediated by the SOX9 destabilization effect or by other HuR targets.

Discussion section: although not experimentally tested, Authors should discuss the specificity of this putative regulatory mechanism in other breast cancer types (luminal, HER2-positive…). Moreover, they should discuss the results in the context of their impact for the EMT phenotype, taking also into account existing literature.

Comments on the Quality of English Language

Only small typos detected that need correction, for instance:

Line 248, “target” not “targets”;

Line 275, “experiment” not “experiments”.

Reviewer 2 Report

Comments and Suggestions for Authors

In this study, Morillo-Bernal and colleagues have identified HuR as an RNA-binding protein capable of binding the {SOX9} mRNA, rendering it stable in triple-negative breast cancer cells, and consequently supporting high levels of expression of SOX9 in these cells.

This is an important investigation that connects two key regulators of protein expression programs in cancer, HuR and SOX9.  The authors need to check with a bit more precision the molecular interaction of HuR with SOX9 mRNA before their model is fully supported, and complement the work with HuR overexpression analysis.  Although the manuscript generally reads well, I encourage the authors to avoid using the word ‘gene’ when they mean ‘mRNA’ or ‘protein’ and have other advice that will reduce ambiguity and improve readability of their paper.

Main comments

1.      In Figure 3, how many sites of HuR interaction are there on the 3’UTR of SOX9 mRNA?  Details about the RNA segment that the authors expressed, biotinylated, and tested in vitro in Figure 3C must be offered in the main paper.  If HuR binds in multiple sites, the authors should test RNA segments that bind HuR and RNA segments that do not bind, to characterize these interactions more comprehensively.

2.      The information on binding must be carried over to Figure 4, where the 3’UTR in the reporter plasmid should be mutated to ablate HuR binding.  The regulation of the reporter chimeric RNA (bearing the luciferase coding region and the full length 3’UTR of SOX9 mRNA) should be evaluated relative to a reporter chimeric RNA in which the 3’UTR lacks HuR binding sites.  Only then can the authors be more certain that HuR directly affects the stability of the reporter.  This is important information, because HuR affects the levels of many regulatory molecules that could indirectly drive SOX9 mRNA levels and stability.  For example, HuR regulates many RBPs (e.g., TTP, which itself can regulate SOX9 mRNA stability) and many microRNAs that could affect SOX9 production.

3.      In Figure 5, the authors show that silencing HuR reduces migration, proliferation, and invasion.  These effects of HuR on cancer cell traits are not themselves new; therefore, the authors need to show evidence that they are linked to the impact of HuR on SOX9 production.  If they overexpress SOX9, can they reverse the effects of HuR on migration, proliferation, invasion?

4.      Although the authors infer that HuR stabilizes the SOX9 mRNA, because its half-life decreases when HuR is silenced, they need to test, at least in several key experiments, if overexpressing HuR increases SOX9 mRNA stability and levels, and SOX9 production.  These experiments ought to be extended to Figure 5, where they should test HuR gain-of-function effects on migration, invasion, and proliferation, and study if these phenotypes are, at least partly, reversed by silencing SOX9.

Comments on the Quality of English Language

1.      The manuscript reads well in general, but the authors should pay detailed attention to the text, particularly around the words ‘gene’ (which should be reserved for the DNA segment), ‘mRNA’ (the expressed transcript), and ‘protein’ (the product of translation).  For example, in the abstract, the phrase “The mRNA half-life and protein level of SOX9 decrease in cells lacking HuR” should read something like “The half-life of {SOX9} mRNA and the levels of SOX9 protein decreased in cells lacking HuR” (where letters in {} are italics).   Similarly In the title, the word ELAVL1 should not be italicized, as it is the official name of HuR.

2.      Line 44: “While the other members are neuronal genes, HuR is…” should read something like: “While other protein members of the ELAVL family are expressed in neurons, HuR is…”

3.      Line 127: for the phrase “Values were normalized to the housekeeping GAPDH gene and …” the authors likely mean something like “Values were normalized to the levels of {GAPDH} mRNA, encoding the housekeeping protein GAPDH and…”.  Again, I encourage the authors not to use ‘gene’ when they are referring to the mRNA.  It is far more accurate and clearer to reserve ‘gene’ for ‘DNA’, despite frequent incorrect use in the literature of the word ‘gene’ to refer to ‘mRNA’.

4.      ‘RIP’, as defined by J.D. Keene and colleagues (who first described the method), stands for ‘ribonucleoprotein immunoprecipitation’ (not ‘RNA immunoprecipitation’).

5.      The term ‘AREs’ is widely considered to be obsolete.  It was coined decades ago when there was only a general notion that decay-promoting sequences often had As and Us.  However, it is now clear that many decay-promoting elements are not AU-rich… many are CU-rich or A-rich or GC-rich, or otherwise lacking AU-rich sequences.  The authors could refer to them simply as ‘regulatory sequences’ on the mRNA.

Reviewer 3 Report

Comments and Suggestions for Authors

Morillo-Bernal et al. investigated the impact of HuR in breast cancer, identifying potential EMT-related targets. Using RNA-Seq on HuR-silenced breast cancer cells, downregulated genes were analyzed. Mapping HuR binding sites revealed 20 genes, including SOX9, known for its dual role in cancer development. Biochemical analyses confirmed HuR's role in stabilizing SOX9 mRNA, and silencing HuR mirrored the migration and invasion phenotypes seen in SOX9-silenced cells in basal-like triple-negative breast cancer.

Overall, the experiments were well designed. The findings were well presented and point to numerous future directions in the realms of HuR studies and breast cancer biology. However, there are a few areas in which the current manuscript could be improved:

1.     It would be nice to see the finding that HuR Binds to SOX9 mRNA is reproducible by repeating the RIP assay in more than just one cell line (e.g., BT549).

2.     It is well-known that SOX9 is essential for triple-negative breast cancer cell survival and metastasis, so the novelty is compromised by focusing solely on SOX9. It is worth investigating what downstream of HuR-SOX9 is impacted in this context to shed light on more potential therapeutic targets. For example, perhaps running ATAC-seq in shHuR-1&2 cells to see what regions on the chromatin become more open upon SOX9 downregulation, and its potential mechanism.

3.     There were only in vitro assays (MTT, migration and invasion assays) to demonstrate the functionality of HuR-SOX9 and implication in breast cancer development. It would strengthen the claim by incorporating some in vivo studies (xenografting in immune-compromised mice, etc.).

4.     Is there any publicly available clinical data set to correlate HuR-SOX and prognosis in breast cancer patients?

Round 2

Reviewer 1 Report

Comments and Suggestions for Authors

The Reviewer acknowledges Authors for the clear responses and the additional information provided. Still, some of the previously raised points were not convincingly answered and need further clarification:

Point 4: The rationale of this experiment suggested by the reviewer is not clear to us. As we state in the text (line 240‐242 of the revised version) “none of the classic EMT‐TFs (SNAI1, SNAI2, TCF3, TWIST1, ZEB1 and ZEB2) appeared to be deregulated in shHuR cells (Dataset S1)”. Only the expression of E-cadherin gene appears to be slightly upregulated after HuR ablation, but we do not believe that this minor increase in E-cadherin gene expression can be detected in WB.

This experiment is designed to see if there might be changes at the protein level of these EMT-related targets that are not observed at the mRNA level in the RNA-seq due to HuR-mediated translational control, reinforcing HuR`s phenotypic effect. Authors should provide WB validation for some of them;

Point 5: We respectfully disagree with the reviewer on this point. Using an extended sequence may result in the binding of HuR to the synthetic RNA in the surrounding sequences instead of the ARE sequence. The selection of the length of biotinylated RNAs was based on previous reports (doi: 10.1128/mcb.13.6.3494‐3504; doi: 10.1073/pnas.91.23.11207;doi: 10.1038/s41467‐020‐ 19664‐2).

That was the precise point of the initial comment. To address if HuR binds or not adjacent sequences. Authors could test at least two equally short sequences from the SOX9 3`UTR (adjacent to but not containing the ARE sequence) to understand if HuR is only binding to the ARE sequence or to the other predicted HuR binding sites (as there are 27). This would clarify if the ARE sequence is the sole determinant of HuR binding;

Point 6: As suggested by the reviewer, we have attempted to conduct this experiment, infecting HuR silenced cells in two distinct genetic backgrounds (Hs578T and BT549) with a lentiviral vector expressing SOX9. In both instances, the cells exhibited mortality, indicating that under these conditions, overexpression of SOX9 is deleterious.

Understanding that the experiment may be technically challenging, without formal demonstration of a rescue effect (even if only partially), Authors cannot sustain the claim that the observed effects are mediated by SOX9, despite observing phenotypes that are similar to the ones of SOX9-silenced cells. Authors carefully avoid this in the Abstract and Introduction sections. But sentences like the one in line 382 “HuR stabilizes SOX9 mRNA to promote cell migration and invasion.” or in line 420 “SOX9 contributes to HuR-silencing dependent effects on migration and invasion.” have to be rephrased as it is not demonstrated. Furthermore, the title of the article also needs to be changed to read something like “HuR (ELAVL1) promotes migration and invasion in breast cancer cells and targets SOX9 mRNA”.

Reviewer 2 Report

Comments and Suggestions for Authors

I appreciate the authors' responsiveness to all the requested revisions.  

Comments on the Quality of English Language

With the requested revisions, the manuscript reads better.

Round 3

Reviewer 1 Report

Comments and Suggestions for Authors

Authors have provided additional information and introduced the necessary modifications in reply to the Reviewer comments.

A minor suggestion regarding the title, it should read "HuR (ELAVL1) Stabilizes SOX9 mRNA and Promotes Migration and Invasion of Breast Cancer Cells".